# Right Cortical Infarction and a Reduction in Putamen Volume May Be Correlated with Empathy in Patients after Subacute Ischemic Stroke—A Multimodal Magnetic Resonance Imaging Study

**DOI:** 10.3390/jcm11154479

**Published:** 2022-07-31

**Authors:** Jian-Feng Qu, Yue-Qiong Zhou, Jian-Fei Liu, Hui-Hong Hu, Wei-Yang Cheng, Zhi-Hao Lu, Lin Shi, Yi-Shan Luo, Lei Zhao, Yang-Kun Chen

**Affiliations:** 1Department of Neurology, Dongguan People’s Hospital (Affiliated Dongguan Hospital, Southern Medical University), Dongguan 523108, China; geoffrey-197830@163.com (J.-F.Q.); yueqiong_zhou@163.com (Y.-Q.Z.); liujianfei1004@163.com (J.-F.L.); hui137344@163.com (H.-H.H.); weiyang_cheng@163.com (W.-Y.C.); zh_lu93@163.com (Z.-H.L.); 2Faculty of Neurology, Graduate School of Guangdong Medical University, Zhanjiang 524013, China; 3Faculty of Neurology, Graduate School of Southern Medical University, Guangzhou 510505, China; 4Department of Imaging and Interventional Radiology, The Chinese University of Hong Kong, Shatin, Hong Kong, China; shilin@cuhk.edu.hk; 5BrainNow Research Institute, Shenzhen 518000, China; lys84818@hotmail.com (Y.-S.L.); zhaolei@link.cuhk.edu.hk (L.Z.)

**Keywords:** subacute ischemic stroke, multimodal magnetic resonance imaging, brain structure volumetry, white matter integrity, empathy

## Abstract

Empathy has not been well studied in patients following ischemic stroke. We aimed to evaluate the relationships of multimodal neuroimaging parameters with the impairment of empathy in patients who had experienced subacute ischemic stroke. Patients who had experienced a first-event acute ischemic stroke were recruited, and we assessed their empathy using the Chinese version of the Empathy Quotient (EQ) 3 months after the index stroke. Multimodal magnetic resonance imaging (MRI) was conducted in all the participants to identify acute infarction and assess brain volumes, white matter integrity, and other preexisting abnormalities. We quantified the brain volumes of various subcortical structures, the ventricles, and cortical lobar atrophy. The microstructural integrity of the white matter was reflected in the mean fractional anisotropy (FA) and mean diffusivity (MD), and the regional mean values of FA and MD were quantified after mapping using the ICBM_DTI_81 Atlas. Twenty-three (56.1%) men and 18 (43.9%) women (mean age: 61.73 years, range: 41–77 years) were included. The median National Institutes of Health Stroke Scale (NIHSS) score at discharge was 1 (range: 0–4). On univariate analysis, the EQ was correlated with right cortical infarction (r = −0.39, *p* = 0.012), putamen volume (r = 0.382, *p* = 0.014), right putamen volume (r = 0.338, *p* = 0.031), and the FA value of the right sagittal stratum. EQ did not correlated with the MD value in any region of interest or pre-existing brain abnormalities. Multiple stepwise linear regression models were used to identify factors associated with EQ. After adjusting for age and the NIHSS score on admission, the frequency of right cortical infarcts negatively correlated with EQ (standardized β = −0.358, 95% confidence interval =−0.708 to −0.076, *p* = 0.016), and the putamen volume positively correlated with EQ (standardized β = 0.328, 95% confidence interval =0.044 to 0.676, *p* = 0.027). In conclusion, in patients who have experienced subacute ischemic stroke, right cortical infarction and a smaller putamen volume are associated with the impairment of empathy.

## 1. Introduction

Social cognition broadly refers to the cognitive processes used to encode and decode social information. It is also critical for people to understand themselves and others, as well as the norms and procedures of the social world [1]. One of the core concerns in social cognition is empathy [2]. Empathy refers to a person’s emotional response to the perceived situation of another [3]; it is an ability to understand the perspective of another person, and it provides an important foundation for relationships, communication, negotiations, and other social activities [4]. Empathy plays important roles in prosocial behavior, the inhibition of aggression, and moral reasoning, by permitting someone to understand another person’s psychological state and experience their emotions [5]. Several neurological and neuropsychiatric diseases disrupt empathy, such as autism, frontotemporal dementia, traumatic head injury, schizophrenia, and stroke [2].

Patients who experience stroke can suffer from a decline in empathy, which has clinical significance [6,7]. Patients who experience emotion recognition deficit may appear to have a variety of interpersonal difficulties, such as complaints of frustration in social relationships, feelings of social discomfort, desire to connect with others, feelings of social disconnection, and the use of controlling behaviors during normal living [6]. Yeh et al. found that, following stroke, the cognitive and emotional empathy of patients are commonly impaired, and emotional processing ability is also affected, leading to a poor outcome [8]. Another study showed that an impairment in social cognition, including empathy, persists in patients with good limb function over the long term following stroke, which leads to a restriction in participation in normal life [9]. Some previous studies have aimed to identify the brain regions engaged in empathy, and showed that specific regions are involved in emotional dysfunction. Leigh et al. found that an acute impairment of empathy is particularly associated with infarcts in the temporal pole and anterior insula [4]. A study that recruited patients following stroke showed that the right posterior superior temporal gyrus in the right hemisphere ventral stream is critical for the identification of emotion in speech [10]. This finding suggests that further assessment of impairment in patients who experience stroke and have damage to this area is required. Another study used functional magnetic resonance imaging (MRI) to show that the bilateral insula can be considered as forming a core network in empathy [11]. In addition, this study showed that cognitive–evaluative and affective–perceptual empathy can be distinguished at the level of regional activation. These findings are consistent with distinct regions being involved in the impairment of empathy. However, current knowledge of the emotional disorders that occur in patients who experience ischemic stroke is limited.

With the development of MRI methods and intelligent image quantification technology, we may be able to better understand the underlying pathogenesis of defects in empathy by combining studies of brain structure and white matter integrity. In the present study, we aimed to evaluate the relationship between the severity of defects in empathy and neuroimaging-derived parameters, including indices of brain structure and white matter integrity, in patients who had experienced subacute ischemic stroke.

## 2. Methods

### 2.1. Participants and Setting

The participants in the study were prospectively recruited at Division I, Department of Neurology, Dongguan People’s Hospital between 1 July 2021 and 30 December 2021. The inclusion criteria were as follows: (1) age > 18 years; (2) a first acute ischemic stroke occurring within the 7 days preceding admission; (3) a complete brain MRI examination was performed at the acute stage, involving T1-weighted (T1W), T2W, and diffusion-weighted imaging (DWI); (4) National Institute of Health stroke scale (NIHSS) score ≤ 15 on admission; and (5) a modified Rankin Scale (mRS) score ≤ 2 on discharge. The exclusion criteria were as follows: (1) a transient ischemic attack occurred; (2) there was a lack of complete MRI; (3) no infarction was detected on DWI; (4) the complication of hemorrhagic transformation occurred; (5) the patient died during hospitalization; (6) severe comorbidities were present (e.g., malignant tumors and, severe organ dysfunction); (7) there was a history of dementia or mental disorder, or obvious cognitive dysfunction or severe depression on admission, identified using medical records; and (8) the patient or their relative refused to provide written informed consent.

The study was performed in accordance with the Declaration of Helsinki, and the study protocol was approved by the Ethics Committee of Dongguan People’s Hospital. Written informed consent was obtained from all the participants.

### 2.2. Collection of Demographic Data

Demographic and clinical data were recorded. The ischemic stroke subtype was determined in accordance with the Trial of Org 10,172 in the Acute Stroke Treatment subtype system by the attending neurologist during hospitalization [12].

### 2.3. Follow-Up Examinations

All the participants were followed up for 3 months after the index stroke via a face-to-face interview. Empathy was assessed using the Chinese version of the Empathy Quotient (EQ) [13], which was translated from the original EQ and validated [13]. The EQ comprises 60 questions, which are divided into two types as follows: 40 questions regarding empathy and 20 filler items. The 20 filler items (items 2, 3, 5, 7, 9, 13, 16, 17, 20, 23, 24, 30, 31, 33, 40, 45, 47, 51, 53, and 56) are included to distract the participant from a relentless focus on empathy. An initial attempt to separate the items into purely affective and cognitive categories was abandoned because in most instances of empathy, the affective and cognitive components are both present and cannot be easily disentangled. To avoid a response bias, approximately half of the items were worded to produce a “disagree” response and half to produce an “agree” in individuals with intact empathy. For the questions used to assess empathy, “definitely agree” responses scored 2 points and “slightly agree” responses scored 1 point for the following items: 1, 6, 19, 22, 25, 26, 35, 36, 37, 38, 41, 42, 43, 44, 52, 54, 55, 57, 58, 59, and 60. Similarly, for questions aimed to elicit negative responses, “definitely disagree” responses scored 2 points and “slightly disagree” responses scored 1 point for the following items: 4, 8, 10, 11, 12, 14, 15, 18, 21, 27, 28,29, 32, 34, 39, 46, 48, 49, and 50. The total EQ score ranges between 0 and 80, with higher scores reflecting greater empathy [14].

To fully characterize the cognitive and mood statuses, in addition to empathy, we also assessed the quality of life using the stroke-specific quality of life scale [15], anxiety using the Hamilton Anxiety Rating Scale [16], depression using the Hamilton Depression Rating Scale [16], and cognition status using the Montreal Cognitive Assessment [17] at the follow-up examination. The NIHSS score, mRS score, recurrence of stroke and mortality rate during the follow-up were also recorded.

### 2.4. MRI Protocol and Analysis

All the participants underwent MRI scans using a 3.0-T scanner (Skyra; Siemens, Erlangen, Germany) on admission and at follow-up examinations. The MRI protocol included T1W, T2W, and DWI at the acute stage and three-dimensional T1W, T2W, and diffusion tensor imaging (DTI) at follow-up.

In MRI during the acute phase, DWI spin echo planar imaging was performed using the following parameters: repetition time (TR)/echo time (TE)/excitation = 2162/76/1, matrix = 128 × 128, field of view (FOV) = 230 mm, slice thickness/gap = 6 mm/1 mm, echo planar imaging factor = 47, and acquisition time = 25.9 s. Three orthogonally applied gradients were used, with b-values of 0 and 1000. Axial SE T1 TR/TE/excitation = 488/15/1, FOV = 230 mm, slice thickness/gap = 6 mm/1 mm, matrix = 256 × 256, and acquisition time = 1 min and 24.8 s and TSE T2 (TR/TE/excitation = 3992/110/2, turbo factor = 15, FOV = 230 mm, slice thickness/gap = 6 mm/1 mm, matrix = 512 × 512); and acquisition time = 1 min and 55.8 s images were also acquired.

For the multimodal MRI at follow-up, T1W was performed in the sagittal plane using a three-dimensional volumetric sequence and the following parameters: TR/TE/inversion time (TI) = 2300/2.27/900 ms, FOV = 250 × 235 mm^2^, matrix = 256 × 240, flip angle = 8°, slice thickness = 1 mm, and inter-slice spacing = 1 mm. DTI MRI was scanned using TR/TE = 9100/100 ms and, matrix = 224 × 224. The slice thickness was 2 mm, the inter-slice spacing was 2.6 mm, and the FOV was =285 × 285 mm^2^. Thirty diffusion gradient directions were realized using a b-value of 1000 s/mm^2^ and a non-DWI volume (b-value = 0 s/mm^2^; number of excitations = 2).

The magnetic resonance images were analyzed as follows. (1) Identification and localization of acute infarction: we visually assessed the site and volume of acute lesions in the DWI sequence. The locations of acute infarct were visually divided into cortical, subcortical, and infratentorial regions [18]. We also recorded unilateral acute cortical infarcts. (2) Brain volumetry: all the follow-up MRI scans were analyzed using AccuBrain^®^ (BrainNow Medical Technology Limited, Shenzhen, China). AccuBrain is a fully automatic neuroanatomical volumetry tool that quantifies the brain volumes of various subcortical structures, ventricles, and cortical lobar atrophy within a clinically acceptable time. AccuBrain performs brain structure segmentation on the basis of a multi-atlas image registration scheme [19]. (3) White matter integrity: the microstructural integrity of the white matter was analyzed using DTI. The resampled data were used to compute the mean fractional anisotropy (FA) and mean diffusivity (MD), and the regional mean values of FA and MD were quantified after mapping using the ICBM_DTI_81 Atlas. Lower values of global FA and higher values of global MD reflected decreased WM microstructural integrity [20]. (4) Brain lesion quantification: we also assessed other brain abnormalities, such as the severity of defects in cortical cholinergic pathways according to Cholinergic Pathways Hyperintensities Scale [21], silent brain infarcts, enlarged perivascular spaces and the volumes of white matter hyperintensitiy. We classified silent brain infarction (SBI) lesions (historic stroke) according to their location, including in the brainstem, cerebellum, basal ganglia, thalamus, and lobes.

### 2.5. Statistical Analysis

Statistical analyses were performed using IBM SPSS for Windows (V 24.0, IBM Corp., Armonk, NY, USA). Descriptive data are presented as proportions, means or medians as appropriate. Univariate correlation analyses of variables with EQ were performed using Pearson’s (normally distributed data) or Spearman’s correlation (categorical or non-normally distributed data) analyses. Variables that were significantly associated with EQ (with *p* < 0.05) were then entered as independent variables into separate multiple linear regression analyses, using a backward stepwise selection strategy, with EQ as the dependent variable. The possibility of multicollinearity was tested for among the independent variables and was accepted when the correlation coefficient was ≥0.6. The significance level was set at 0.05 (two-sided).

## 3. Results

Two hundred and eighty-six patients who had experienced a first acute ischemic stroke were consecutively admitted during the study period. The selection of patients is shown in a flow chart (Figure 1). Forty-one patients were included in the final analysis. A comparison between the excluded and included patients showed no differences in age (59.9 ± 14.9 years vs. 61.7 ± 9 years, respectively, *p* = 0.464) or sex (men, 67.9% vs. 56.1%, respectively, *p* = 0.143), but there were significant differences in the NIHSS score on admission (4 (2–8) vs. 2 (0–3), respectively, *p* < 0.001) and the length of stay in hospital (12 (8–17) days vs. 6 (5–9) days, respectively, *p* < 0.001).

The baseline characteristics of the participants are shown in Table 1. The study sample consisted of 23 (56.1%) men and 18 (43.9%) women, with a mean age of 61.73 years (range, 41–77 years). The median NIHSS score at discharge was 1 (range: 0–4).

The results of the other assessments at follow-up were as follows: SSQOL: 220.2 ± 21.6, HAMA: 3 (2–7), HAMD: 5 (2–10.5), MoCA: 22 (18.5–24.5), NIHSS: score: 0 (0–1), and mRS: 1 (0–1).

### 3.1. Univariate Correlates of EQ at Follow-Up

The univariate analyses are shown in Table 2, Table 3, Table 4, Table 5 and Table 6. It revealed that EQ was found to correlate with right cortical infarction (r = −0.39, *p* = 0.012), putamen volume (r = 0.382, *p* = 0.014), right putamen volume (r = 0.338, *p* = 0.031), and the FA value of the right sagittal stratum. EQ did not correlated with either the MD values of the regions of interest, or pre-existing brain abnormalities, including the location of SBI.

### 3.2. Linear Regression Analysis of the EQ

The multiple stepwise linear regression models are shown in Table 7. Right frontal lobe infarct was not included in the model because it strongly correlated with right cortical infarction (r = 0.668). After adjustment for age and the NIHSS score on admission, the right putamen volume and FA of the right sagittal stratum did not significantly correlate with the EQ. The presence of a right cortical infarcts, significantly negatively correlated with the EQ, with a standardized β of −0.358 (95% confidence interval = −0.708 to −0.076, *p* = 0.016). Putamen volume was positively correlated with EQ, with a standardized β of 0.328 (95% confidence interval = 0.044 to 0.676, *p* = 0.027).

## 4. Discussions

By means of this observational study, we aimed to evaluate the relationships of biomarkers obtained by neuroimaging using multimodal MRI with the impairment of empathy in patients who had experienced subacute ischemic stroke. We assessed the empathy status of the patients 3 months after the index stroke. Brain structure volumetry of the various subcortical and ventricular structures and major lobar atrophy were quantified at follow-up using three-dimensional T1W images. In addition, neuroimaging-derived markers of white matter integrity, such as mean FA and mean MD, of the regions of interest defined in the ICBM_DTI_81 Atlas, were quantified. The principal findings were that the frequency of right cortical infarcts and the putamen volume significantly correlate with empathy in patients who have experienced subacute ischemic stroke.

In the present study, the frequency of right cortical infarcts, significantly negatively correlated with EQ (Figure 2), with a standardized β value of −0.358. This finding suggests that the presence of right cortical infarcts is associated with a lower EQ, indicating poorer empathy. Ischemic brain injury can lead to the impairment of empathy. Leigh et al. studied 27 patients with acute right hemisphere ischemic stroke and 24 neurologically intact inpatients using a test of affective empathy [4], and found that an acute impairment of affective empathy is associated with infarcts in the temporal pole and anterior insula. Another study showed that stroke involving the right posterior superior temporal gyrus in the right hemisphere ventral stream was critical for the identification of emotion in speech [10]. This finding suggests that patients who have experienced stroke and infarction in this area should be assessed for the impairment of emotion. Although these studies were inconclusive, they suggest that a network may play a major role in the processing of empathy. The present findings are consistent with the notion that the right cortical region may play an important role in the recognition of emotion.

In the present study, we also assessed the relationship between brain structure volumetry and empathy in patients who had experienced acute ischemic stroke. We found that the putamen volume significantly positively correlated with the EQ (Figure 3), with a standardized β value of 0.328. We also obtained data regarding putaminal volumes using quantitative images and present the corresponding relationships of EQ for each participant in the Appendix A. These findings show that a smaller putamen volume is associated with a more severe impairment of empathy. As a sequela of stroke may be related to the index stroke and/or SBIs, we assessed the relationship between EQ and the location of SBI (brainstem, cerebellum, basal ganglia, thalamus, or lobes), and found that the location of SBI was not significantly associated with EQ. This implies that a smaller putamen may be more affected both by a combination of stroke and a constitutional condition than by a single factor.

Some recent neuroimaging studies have identified a series of brain regions that are involved in empathy. A functional MRI study showed that stimuli are associated with greater activations of the caudate-putamen, paracingulate, anterior and posterior cingulate, and amygdala than that predicted by the theory of mind stimuli [22]. The authors of another study concluded that the dorsal anterior cingulate cortex-anterior mid-cingulate cortex-supplementary motor area and the bilateral insula can be considered to form a core network in empathy [23]. In additional, they concluded that cognitive evaluative and affective–perceptual empathy can be distinguished at the level of regional activation. The putamen is an important part of the lateral pathway of the cortical cholinergic pathway [24]. It is a subcortical structure that forms part of the dorsal striatum of the basal ganglia, and is conventionally thought to be associated with the reinforcement of learning and motor control, including speech articulation [25]. A study of macaque monkeys showed that deep and superficial cortical white matter neurons, peri-claustral white matter neurons, and the claustrum proper project to the putamen [26]. Moreover, Zhao et al. showed that the putamen may play an important role in instrumental learning processes, and that structural variations of associated brain regions are required for the successful acquisition of functional MRI neurofeedback-guided self-regulation [27]. The involvement of the unilateral putamen in various language components suggests that it may play a role in language [28]. A smaller putamen volume indicates the possibility of injury to neurons and projecting fibers, which may interfere with the processing of language during learning and the recognition of emotion, and thus be involved in empathy.

The present study had the following strengths. (1) We assessed the patients’ empathy after subacute ischemic stroke. (2) We used a comprehensive set of neuroimaging parameters obtained using a multimodal MRI protocol, including acute infarction, brain volumetry, white matter integrity, and other preexisting abnormalities. There were also some limitations to the study. First, the sample size was relatively small. Second, other social cognition domains were not evaluated, such as the theory of mind, perception and behavior. Third, the recruited patients had experienced relatively mild stroke, which may limit the generalizability of the current findings. Fourth, because the range of stroke subtypes was relatively limited, with only 5% of participants having atrial fibrillation, the representativeness of the sample may be insufficient to generalize the findings to all patients who experience stroke.

In conclusion, in patients with subacute ischemic stroke, right cortical infarction is negatively correlates with empathy, while the putamen volume is positively correlates with empathy. If these findings can be confirmed in a larger study, they should contribute to the understanding of the loss of empathy that occurs following stroke.

## Figures and Tables

**Figure 1 jcm-11-04479-f001:**
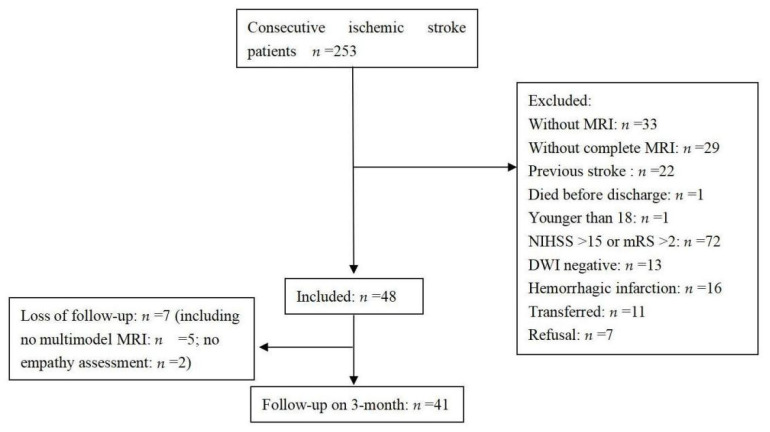
Flow-chart of the study. NIHSS: National Institutes of Health Stroke Scale, DWI: Diffusio-weighted imaging, MRI: magnetic resonance imaging.

**Figure 2 jcm-11-04479-f002:**
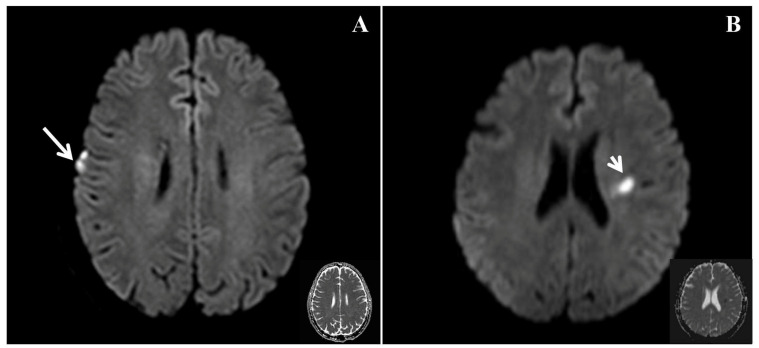
Diffusio-weighted imaging (DWI) and the corresponding apparent diffusion coefficient (ADC) in patients with differing empathy statuses. (**A**) A patient with an EQ of 16 (poorer empathy) had an acute infarct involving the right frontal lobe marked by a long arrow (right cortical infarct) during the acute phase. (**B**) A patient with an EQ of 73 (better empathy) had an acute infarct in the left corona radiata marked by a short arrow (left subcortical infarct) during the acute phase.

**Figure 3 jcm-11-04479-f003:**
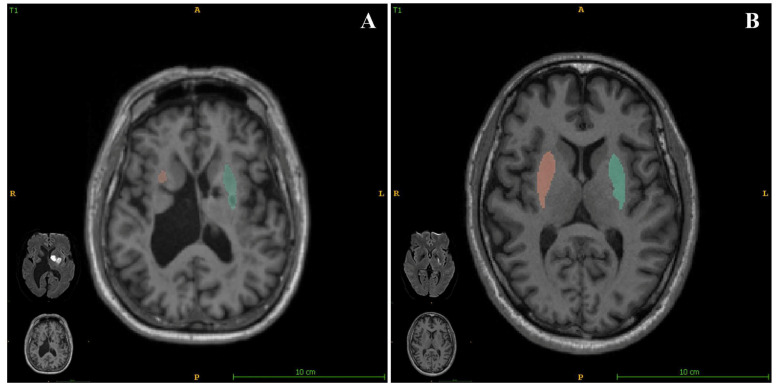
AccuBrain^®^ (BrainNow Research Institute, Shenzhen, China) was used to quantify of the volume of the putamen and corresponding diffusion-weighted imaging (DWI) in patients with differing empathy statuses. (**A**) Image of a patient with an EQ of 23 (poorer empathy), with a total putamen volume of 5.53 mL. (**B**) Image of a patient with EQ of 57 (better empathy) with a total putamen volume of 11.6 mL.

**Table 1 jcm-11-04479-t001:** Results of the correlation analysis of the relationships between empathy and clinical characteristics.

Variables	Mean (SD)/Median (IQR)/*n* (%) *n* = 41	EQ	
		r	*p*
Age (years)	61.73 ± 9.02	−0.129	0.421
Men	23 (56.1%)	0.15	0.35
Hypertension	33 (80.5%)	0.198	0.215
Diabetes mellitus	11 (26.8%)	0.051	0.75
Atrial fibrillation	2 (4.9%)	−0.249	0.116
NIHSS on admission	2 (0–3)	−0.011	−0.947
NIHSS on discharge	1 (0–1)	−0.059	0.714
TOAST		0.023	0.886
LAA	16 (39%)		
SAO	3 (7.3%)		
CE	17 (41.5%)		
SOC	1 (2.4%)		
SUC	4 (9.8%)		
Antiplatelet drugs	40 (97.6%)	0.147	0.359
Endovascular therapy	3 (7.3%)	−0.059	0.712
Delirium	1 (2.4%)	−0.201	0.209
Sedatives	1 (2.4%)	−0.201	0.209
Anti-psychotics	1 (2.4%)	−0.201	0.209
Length of hospitalized day	6 (5–9)	−0.27	0.088

EQ: empathy quotient, LAA: large artery atherosclerosis, SAO: small artery occlusion, CE: cardioembolism, SOC: stroke with another cause, SUC: stroke of unknown cause, L: left, R: right, IQR: interquartile range.

**Table 2 jcm-11-04479-t002:** Results of correlation analysis for the relationships between empathy and structural magnetic resonance imaging-derived parameters.

Variables	Mean (SD)/Median (IQR)/*n* (%) *n* = 41	EQ	
		r	*p*
Location of Infarcts			
L-cortical	10 (24.4%)	−0.127	0.428
R-cortical	8 (19.5%)	−0.39	0.012
L-subcortical	17 (41.5%)	0.128	0.426
R-subcortical	14 (34.1%)	−0.033	0.84
Infratentorial	8 (19.5%)	0.029	0.859
L-frontal lobe	6 (14.6%)	−0.274	0.083
R-frontal lobe	4 (9.8%)	−0.375	0.016
L-temporal lobe	4 (9.8%)	−0.209	0.191
R-temporal lobe	4 (9.8%)	−0.236	0.137
L-parietal lobe	5 (12.2%)	−0.303	0.054
R-parietal lobe	4 (9.8%)	−0.181	0.258
L-occipital lobe	2 (4.9%)	−0.273	0.084
R-occipital lobe	2 (4.9%)	−0.148	0.354
Infarct volume	0.92 (0.48–2.33)	−0.273	0.084
SBI	14 (9–21.5)	0.185	0.246
EPVS	1 (1–3)	0.049	0.762
CHIPS	52 ± 14.4	−0.024	0.882
WMH volume	4.25 (1.94–13.1)	0.087	0.587
PVH volume	3.57 (1.095–11.15)	0.102	0.527
DWMH volume	0.838 (0.349–1.45)	−0.176	0.272

EQ: Empathy quotient, L: left, R: right, SBI: silent brain infarcts, EPVS: enlarged perivascular space, CHIPS: Cholinergic Pathways Hyperintensities Scale, WMH: white matter hyperintensity, PVH: periventricular hyperintensity, DWMH: deep white matter hyperintensity. The bold in the table footer represented for significant correlation and *p* < 0.05.

**Table 3 jcm-11-04479-t003:** Results of correlation analysis for the relationships between empathy and brain structure total volumes.

Brain Structure	Volume	EQ	
		r	*p*
ICV	1426.6 ± 121.44	0.015	0.926
Brain parenchyma	1102.71 ± 99.25	0.105	0.512
Hippocampus	6.8 ± 0.7	0.169	0.289
Amygdala	3.67 ± 0.37	0.068	0.673
Thalamus	11.36 ± 1.91	0.264	0.095
Caudate	6.84 ± 0.97	0.099	0.537
Putamen	10.4 ± 1.41	**0.322**	**0.040**
Pallidum	3.03 ± 0.47	0.131	0.414
Accumbens	1.1 ± 0.14	0.043	0.792
Hypothalamus	0.68 ± 0.09	−0.128	0.426
Midbrain	5.79 ± 0.623	−0.006	0.969
Pons	13.41 ± 1.97	−0.017	0.918
Medulla	4.11 ± 0.42	0.051	0.749
SCP	0.22 ± 0.03	−0.159	0.321
Cerebellum	130.73 ± 12.51	−0.202	0.205

EQ: Empathy quotient, ICV: intracranial volume, SCP: superior cerebellar peduncle. The bold in the table footer represented for significant correlation and *p* < 0.05.

**Table 4 jcm-11-04479-t004:** Results of the correlation analysis for the relationships between empathy and unilateral brain volumetry.

Brain Structure	Volume (mL)	EQ	
		r	*p*
Hippocampus-L	3.31 ± 0.37	0.081	0.614
HIppocampus R	3.48 ± 0.39	0.222	0.164
Amygdala-L	1.84 ± 0.2	0.044	0.783
Amygdala-R	1.84 ± 0.19	0.087	0.588
Lateral ventricle-L	16.58 ± 13.09	−0.044	0.784
Lateral ventricle-R	14.43 ± 12.81	−0.120	0.456
Inf-Lat-Vent-L	1.65 ± 0.79	−0.134	0.404
Inf-Lat-Vent-R	1.59 ± 0.8	−0.098	0.541
Thalamus-L	5.79 ± 0.49	0.250	0.115
Thalamus-R	5.82 ± 0.7	0.307	0.051
Caudate-L	3.46 ± 0.41	0.047	0.772
Caudate-R	3.45 ± 0.77	−0.039	0.807
Putamen-L	5.07 ± 0.67	0.254	0.109
Putamen-R	5.33 ± 0.8	**0.338**	**0.031**
Pallidum-L	1.56 ± 0.27	−0.004	0.981
Pallidum-R	1.47 ± 0.23	0.273	0.085
Accumbens-L	0.55 ± 0.08	0.065	0.685
Accumbens-R	0.55 ± 0.07	0.048	0.767
Hypothalamus-L	0.33 ± 0.04	−0.084	0.603
Hypothalamus-R	0.36 ± 0.04	−0.085	0.597

EQ: Empathy quotient, L: left, R: right, Inf-Lat-Ven: inferior angle of the lateral ventricle. The bold in the table footer represented for significant correlation and *p* < 0.05.

**Table 5 jcm-11-04479-t005:** Results of correlation analysis for the relationships between empathy and brain atrophy.

Brain Structure	Atrophy Ratio (%)	EQ	
		r	*p*
Frontal lobe-L	36.52 ± 7.13	−0.008	0.959
Frontal lobe-R	35 ± 60.7	−0.037	0.818
Occipital lobe-L	13 ± 3.37	−0.269	0.089
Occipital lobe-R	9.7 ± 2.88	−0.269	0.089
Temporal lobe-L	29.5 ± 5.91	−0.251	0.113
Temporal lobe-R	20.46 ± 3.02	−0.026	0.870
Parietal lobe-L	38.42 ± 8.22	−0.064	0.693
Parietal lobe-R	33.9 ± 8.06	−0.112	0.485
Cingulate-L	9.76 ± 3.49	−0.227	0.154
Cingulate-R	14.12 ± 3.18	−0.260	0.101
Insular-L	28.58 ± 15.7	−0.050	0.755
Insular-R	20.9 ± 9.88	−0.046	0.777
Cerebellum	9.66 ± 2.71	0.029	0.857
MTLA-L	0.51 ± 0.28	−0.125	0.436
MTLA-R	0.47 ± 0.27	−0.130	0.416

EQ: Empathy quotient, L: left, R: right, MTLA: medial temporal lobe atrophy.

**Table 6 jcm-11-04479-t006:** Results of the correlation analysis for the relationships between empathy and white matter integrity.

Brain Structure	MD	EQ		FA	EQ	
		r	*p*		r	*p*
MCP	**0.74 (0.72–0.78)**	0.048	0.764	0.57 (0.55–0.59)	−0.102	0.525
PCT	0.75 (0.71–0.82)	0.147	0.358	0.51 (0.47–0.54)	−0.127	0.428
GCC	0.83 (0.79–0.91)	−0.080	0.617	0.59 (0.56–0.62)	0.055	0.734
BCC	0.94 (0.87–1.04)	−0.148	0.357	0.59 (0.54–0.64)	0.186	0.244
SCC	0.9 (0.86–0.97)	−0.099	0.538	0.67 (0.63–0.69)	0.129	0.423
FX	1.63 (1.36–1.85)	−0.130	0.419	0.45 (0.41–0.56)	0.085	0.595
CST-R	0.74 (0.72–0.8)	0.184	0.248	0.58 (0.53–0.61)	−0.042	0.795
CST-L	0.77 (0.73–0.81)	−0.007	0.965	0.58 (0.54–0.61)	−0.020	0.900
ML-R	0.75 (0.73–0.8)	−0.022	0.890	0.63 (0.61–0.64)	−0.051	0.752
ML-L	0.76 (0.74–0.8)	−0.111	0.490	0.62 (0.61–0.64)	0.029	0.859
ICP-R	0.75 (0.74–0.78)	−0.187	0.241	0.57 (0.55–0.59)	−0.052	0.746
ICP-L	0.76 (0.74–0.81)	−0.290	0.066	0.58 (0.56–0.59)	0.066	0.682
SCP-R	0.95 (0.9–1.01)	0.018	0.910	0.69 (0.66–0.71)	−0.005	0.976
SCP-L	0.98 (0.91–1.07)	−0.032	0.842	0.68 (0.66–0.71)	0.084	0.603
CP-R	0.79 (0.76–0.83)	−0.179	0.262	0.66 (0.64–0.68)	0.075	0.639
CP-L	0.78 (0.74–0.83)	−0.195	0.222	0.66 (0.62–0.68)	0.077	0.633
ALIC-R	0.77 (0.72–0.82)	0.014	0.929	0.54 (0.51–0.57)	0.000	0.998
ALIC-L	0.78 (0.72–0.84)	0.011	0.945	0.52 (0.48–0.54)	0.073	0.652
PLIC-R	0.72 (0.7–0.75)	0.109	0.498	0.66 (0.63–0.68)	−0.116	0.469
PLIC-L	0.72 (0.7–0.74)	0.151	0.345	0.66 (0.63–0.67)	−0.282	0.074
RLIC-R	0.79 (0.76–0.84)	−0.042	0.796	0.61 (0.57–0.63)	−0.237	0.135
RLIC-L	0.8 (0.74–0.85)	−0.009	0.953	0.63 (0.59–0.65)	0.047	0.772
ACR-R	0.8 (0.76–0.86)	0.028	0.862	0.43 (0.39–0.45)	0.062	0.700
ACR-L	0.78 (0.76–0.87)	0.007	0.966	0.41 (0.36–0.44)	0.014	0.930
SCR-R	0.75 (0.72–0.85)	−0.165	0.303	0.49 (0.43–0.52)	0.147	0.360
SCR-L	0.77 (0.72–0.84)	−0.048	0.768	0.49 (0.44–0.51)	0.066	0.682
PCR-R	0.83 (0.8–0.92)	0.015	0.928	0.48 (0.44–0.51)	−0.070	0.661
PCR-L	0.83 (0.8–0.9)	0.131	0.416	0.47 (0.42–0.5)	−0.022	0.891
PTR-R	0.84 (0.82–0.86)	0.121	0.451	0.6 (0.57–0.62)	−0.116	0.472
PTR-L	0.86 (0.82–0.91)	−0.058	0.721	0.59 (0.56–0.62)	0.011	0.946
SS-R	0.85 (0.81–0.87)	0.188	0.240	0.55 (0.52–0.58)	**−337**	**0.031**
SS-L	0.87 (0.84–0.91)	0.020	0.902	0.55 (0.53–0.57)	−0.186	0.243
EC-R	0.79 (0.75–0.84)	−0.182	0.253	0.45 (0.41–0.47)	−0.056	0.729
EC-L	0.8 (0.77–0.84)	−0.042	0.793	0.45 (0.42–0.47)	0.048	0.764
CGC-R	0.74 (0.72–0.77)	−0.080	0.618	0.5 (0.46–0.52)	0.070	0.663
CGC-L	0.76 (0.72–0.78)	−0.096	0.550	0.5 (0.47–0.53)	0.014	0.931
CGH-R	0.78 (0.76–0.82)	−0.166	0.299	0.48 (0.45–0.51)	−0.134	0.403
CGH-L	0.81 (0.78–0.85)	−0.242	0.128	0.48 (0.46–0.5)	−0.136	0.395
FX/ST-R	1.03 (0.93–1.12)	−0.077	0.632	0.53 (0.47–0.55)	−0.040	0.804
FX/ST-L	0.87 (0.83–0.91)	0.222	0.163	0.55 (0.52–0.58)	−0.025	0.879
SLF-R	0.75 (0.72–0.77)	−0.132	0.410	0.5 (0.47–0.52)	0.081	0.616
SLF-L	0.73 (0.7–0.79)	0.049	0.763	0.5 (0.46–0.52)	0.243	0.126
SFO-R	0.79 (0.73–0.91)	−0.153	0.339	0.47 (0.4–0.51)	0.076	0.639
SFO-L	0.91 (0.75–0.96)	−0.112	0.488	0.44 (0.37–0.47)	0.205	0.198
IFO-R	0.79 (0.77–0.83)	−0.035	0.827	0.54 (0.5–0.55)	0.069	0.670
IFO-L	0.82 (0.77–0.84)	0.003	0.986	0.51 (0.47–0.53)	0.223	0.162
UNC-R	0.77 (0.74–0.8)	0.025	0.879	0.53 (0.49–0.57)	0.104	0.519
UNC-L	0.75 (0.72–0.79)	−0.156	0.329	0.53 (0.51–0.55)	0.207	0.193
TAP-R	1.75 (1.51–1.94)	−0.157	0.326	0.43 (0.37–0.47)	0.073	0.651
TAP-L	1.96 (1.67–2.15)	−0.101	0.529	0.38 (0.33–0.42)	0.136	0.398

MD: mean diffusivity, FA: fractional anisotropy, EQ: Empathy quotient, L: left, R: right, MCP: middle cerebellar peduncle, PCT: pontine crossing tract (part of the MCP); GCC: genu of the corpus callosum, BCC: body of the corpus callosum, SCC: splenium of the corpus callosum, FX: fornix, CST: corticospinal tract, ML: medial lemniscus, ICP: inferior cerebellar peduncle, SCP: superior cerebellar peduncle, CP: cerebral peduncle, ALIC: anterior limb of the internal capsule, PLIC: posterior limb of the internal capsule, RLIC: retrolenticular part of the internal capsule, ACR: anterior corona radiata, SCR: superior corona radiata, PCR: posterior corona radiata, PTR: posterior thalamic radiation (including optic radiation), SS: sagittal stratum (including inferior longitudinal fasciculus and inferior fronto-occipital fasciculus), EC: external capsule, CGC: cingulum (cingulate gyrus), CGH: cingulum (hippocampus), FX/ST: fornix (cres)/stria terminalis (cannot be resolved with the current resolution), SLF: superior longitudinal fasciculus, SFO: superior fronto-occipital fasciculus (may be a part of the anterior internal capsule), IFO: inferior fronto-occipital fasciculus, UNC: uncinate fasciculus, TAP: tapetum. The bold in the table footer represented for significant correlation and *p* < 0.05.

**Table 7 jcm-11-04479-t007:** Results of linear regression analysis.

Variables	EQ (R^2^ = 0.192)	
	*Standardized β*	*p*
Right cortical infarction	−0.358 (−0.708–−0.076)	0.016
Putamen volume	0.328 (0.044–0.676)	0.027
Right putamen volume	−0.114 (−0.336–0.288)	0.854
Right SS-FA	−0.222 (−0.547–0.086)	0.148

The model was adjusted for age and, the NIHSS score on admission, EQ: Empathy Quotient, SS: sagittal stratum, FA: fractional anisotropy.

## Data Availability

The original data generated during the study are included in the article. Further inquiries can be directed to the corresponding author.

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
