# Peer review of "Right Cortical Infarction and a Reduction in Putamen Volume May Be Correlated with Empathy in Patients after Subacute Ischemic Stroke—A Multimodal Magnetic Resonance Imaging Study"

_jcm, 2022, doi:10.3390/jcm11154479_

Round 1

Reviewer 1 Report

his manuscript deals with the limited empathy of patients who have suffered a stroke. This is also where the biggest problem is found: the wording. The authors do not succeed in making it clear to the reader in the text whether, in their opinion, decreasing empathy, increasing empathy, or the same empathy is the problem for these patients. Important preliminary studies are not mentioned (e.g. Guo et al.). For the fact that the authors refer several times to the "prospective data collection", they then have to show missing data for very many patients (e.g. length of stay?). The text also seems erratic and often revised: in the methods section, "severe depression" is used several times as an exclusion criterion (once would be enough).

However, the authors do not dispel an elementary problem: Is the smaller putamen really a stroke sequelae (does not look like that in Fig 3) or do the authors investigate here a constitutional condition?

Only 5% of stroke patients had atrial fibrillation? This suggests an enormous bias due to the pre-selection of the patients - so that the statement of the paper could not be generalized.

Author Response

Responses to Reviewers

We thank the editor and reviewers for their comments. We have revised the manuscript according to their recommendations, as detailed below.

Reviewer #1:

1) His manuscript deals with the limited empathy of patients who have suffered a stroke. This is also where the biggest problem is found: the wording. The authors do not succeed in making it clear to the reader in the text whether, in their opinion, decreasing empathy, increasing empathy, or the same empathy is the problem for these patients. Important preliminary studies are not mentioned (e.g. Guo et al.).

Authors’ reply: We thank the reviewer for their comment. Patients who experience stroke can undergo a decline in empathy that is of clinical significance.1,2 Patients who experience deficits in the recognition of emotion may have a variety of difficulties with their personal relationships, such as complaints of frustration in social relationships, feelings of social discomfort, desire to connect with others, feelings of social disconnection, and the use of controlling behaviors, which are all relevant to normal living.2 We have added this description to the third paragraph of the Introduction on Page 2.

  • For the fact that the authors refer several times to the "prospective data collection", they then have to show missing data for very many patients (e.g. length of stay?).

Authors’ reply: We thank the reviewer for their comment. When the excluded and included patients were compared, there were no differences in age (59.9±14.9 years vs. 61.7±9 years, P=0.464) or sex (men, 67.9% vs. 56.1%, P=0.143), but there were significant differences in NIHSS score on admission (4 [2–8] vs. 2 [0–3], P<0.001) and the length of stay in hospital (12 [8–17] days vs. 6 [5–9] days, P<0.001). We have added this description to the Results on Page 5 and the Discussion on Page 8.

  • The text also seems erratic and often revised: in the methods section, "severe depression" is used several times as an exclusion criterion (once would be enough).

Authors’ reply: We thank the reviewer for their comment. In response, we have revised the Methods text accordingly on Page 3.

  • However, the authors do not dispel an elementary problem: Is the smaller putamen really a stroke sequelae (does not look like that in Fig 3) or do the authors investigate here a constitutional condition?

Authors’ reply: Because a sequela of stroke may be related to the index stroke or silent brain infarct, we have also analyzed the relationship between EQ and the location of SBI (brainstem, cerebellum, basal ganglia, thalamus, or lobes). We found that there was no significant relationship between SBI and EQ. This implies that the relationship between the small size of a brain structure, such as that of the putamen, and EQ, may be affected both by stroke and a constitutional condition, rather than just by a single factor. We have revised Table 2, the Methods section on Page 4, the Results section on Page 6, and the Discussion on Page 7 accordingly.

  • Only 5% of stroke patients had atrial fibrillation? This suggests an enormous bias due to the pre-selection of the patients - so that the statement of the paper could not be generalized.

Authors’ reply: Because the subtypes of stroke experienced by the enrolled patients were relatively limited in number, such that only 5% of patients had atrial fibrillation, the present findings may not be generalizable to every patient that experiences stroke. We have added this as a limitation to the Discussion on Page 8.

Reviewer 2 Report

This study aimed to examine the relationship between empathy and related microstructural integrity of the white matter and brain regional volumes in 41 mild subacute ischemic stroke patients. The authors conclude that subacute ischemic stroke, right cortical infarction and a smaller putamen volume are associated with the impairment of empathy.

Comments:

1.      Page 3, exclusion criteria. “a history of dementia, mental disorder, or severe depression, or obvious cognitive dysfunction or severe depression on admission;” How did you determine cognitive dysfunction or depression? Did the Chinese version of the EQ detect cognitive dysfunction in longitudinal analysis? Baseline cognitive function or mood disorders, even mild change, affects emotional change during the course of hospitalization. Please address cognitive test etc. in the Methods section.

2.      Discussion, lines 219-220. “In this study, right cortical infarcts were significantly negatively correlated with the 219 EQ (Figure 2), with a standardized β value of -0.358.”. The authors present examples of MRI DWI in the right cortical infarction and left subcortical infarction, respectively. To strengthen your findings, you should present data associated with putaminal volumes. Superimposed or other 3D-T1WI images for each patient should be included.

3.      The authors suggested that a smaller putamen volume was correlated with severer impairment of empathy. Why did you use total/bilateral putamen volumes rather than unilateral or mean data? If AccuBrain can calculate these segmentation data, then present them in supplementary data. These are quite important to ascribe Wallerian degeneration and/or relationships of functional connectivity.

4.      Please address treatments such as antiplatelet, endovascular therapy, sedatives and anti-psychotics.

5.      Length of NICU stay and hospital day, and presence of delirium during the hospitalization. These are significantly affects patients’ emotional status and can be included in correlation analysis.

Author Response

Responses to Reviewers

We thank the editor and reviewers for their comments. We have revised the manuscript according to their recommendations, as detailed below.

Reviewer #2

  • Page 3, exclusion criteria. “a history of dementia, mental disorder, or severe depression, or obvious cognitive dysfunction or severe depression on admission;” How did you determine cognitive dysfunction or depression? Did the Chinese version of the EQ detect cognitive dysfunction in longitudinal analysis? Baseline cognitive function or mood disorders, even mild change, affects emotional change during the course of hospitalization. Please address cognitive test etc. in the Methods section.

Authors’ reply: We thank the reviewer for this suggestion. Because the cognitive status and mood of patients are unstable during the acute phase of stroke3, we did not assess these parameters during the period of hospitalization. Therefore, we excluded patients with “a history of dementia, mental disorder, severe depression, or obvious cognitive dysfunction on admission”, principally on the basis of their medical records. We have revised the Methods section on Page 3 to confirm this.

In addition, to better understand the cognitive status and mood of the participants, we assessed anxiety using the Hamilton Anxiety Rating Scale, depression using the Hamilton Depression Rating Scale, and cognitive status using the Montreal Cognitive Assessment at a follow-up examination. We have revised the cognitive tests section of the Methods on Page 4 accordingly.

2) Discussion, lines 219-220. “In this study, right cortical infarcts were significantly negatively correlated with the 219 EQ (Figure 2), with a standardized β value of -0.358.”. The authors present examples of MRI DWI in the right cortical infarction and left subcortical infarction, respectively. To strengthen your findings, you should present data associated with putaminal volumes. Superimposed or other 3D-T1WI images for each patient should be included.

Authors’ reply: We thank the reviewer for this suggestion. We present data regarding the relationships of putaminal volumes, measured using quantitative images, with EQ for each patient as Supplemental materials and we have revised the Discussion on Page 7 accordingly.

3) The authors suggested that a smaller putamen volume was correlated with severer impairment of empathy. Why did you use total/bilateral putamen volumes rather than unilateral or mean data? If AccuBrain can calculate these segmentation data, then present them in supplementary data. These are quite important to ascribe Wallerian degeneration and/or relationships of functional connectivity.

Authors’ reply: We thank the reviewer for this comment. AccuBrain can be used to calculate the total and unilateral volumes of each brain structure. These data are already presented in Tables 3 and 4.

4) Please address treatments such as antiplatelet, endovascular therapy, sedatives and anti-psychotics.

Authors’ reply: We thank the reviewer for this suggestion. In the present study, 40 participants (97.6%) were using antiplatelet drugs, 3 (7.3%) had undergone endovascular therapy, and 1 (2.4%) was using a sedative and an anti-psychotic. In the univariate analysis, EQ was not associated with these variables. We have revised Table 1 to include these data.

5) Length of NICU stay and hospital day, and presence of delirium during the hospitalization. These are significantly affects patients’ emotional status and can be included in correlation analysis.

Authors’ reply: We thank the reviewer for this suggestion. In the present study, the length of the hospital stay was 6 [5–9] days and the prevalence of post-stroke delirium was 2.4% (1 patient experienced delirium during their hospitalization). In the univariate analysis, EQ was not associated with these variables. We have revised Table 1 to include these data.

References

  1. Rajamanickam Yuvaraj, Murugappan Murugappan, Mohamed Ibrahim Norlinah, Kenneth Sundaraj, Mohamad Khairiyah. Review of emotion recognition in stroke patients. Dement Geriatr Cogn Disord . 2013;36(3-4):179-96. doi: 10.1159/000353440.
  2. Xiuyan Guo, Li Zheng, Wei Zhang, Lei Zhu, Jianqi Li, Qianfeng Wang, et al. Empathic neural responses to others' pain depend on monetary reward. Soc Cogn Affect Neurosci . 2012 Jun;7(5):535-41. doi: 10.1093/scan/nsr034.
  3. Milija D Mijajlović, Aleksandra Pavlović, Michael Brainin, Wolf-Dieter Heiss, Terence J Quinn, Hege B Ihle-Hansen, et al. Post-stroke dementia - a comprehensive review. BMC Med . 2017 Jan 18;15(1):11. doi: 10.1186/s12916-017-0779-7.

Round 2

Reviewer 2 Report

None.